# The Role of the NRF2 Pathway in the Pathogenesis of Viral Respiratory Infections

**DOI:** 10.3390/pathogens13010039

**Published:** 2023-12-31

**Authors:** Maria Daskou, Leila Fotooh Abadi, Chandrima Gain, Michael Wong, Eashan Sharma, Arnaud John Kombe Kombe, Ravikanth Nanduri, Theodoros Kelesidis

**Affiliations:** 1Department of Medicine, Division of Infectious Diseases, David Geffen School of Medicine, University of California Los Angeles, Los Angeles, CA 90095, USA; 2Department of Internal Medicine, Division of Infectious Diseases and Geographic Medicine, University of Texas Southwestern Medical Center, Dallas, TX 75390, USA; leila.fotoohabadi@utsouthwestern.edu (L.F.A.); ravikanth.nanduri@utsouthwestern.edu (R.N.)

**Keywords:** NRF2 pathway, respiratory viruses, viral replication, inflammation

## Abstract

In humans, acute and chronic respiratory infections caused by viruses are associated with considerable morbidity and mortality. Respiratory viruses infect airway epithelial cells and induce oxidative stress, yet the exact pathogenesis remains unclear. Oxidative stress activates the transcription factor NRF2, which plays a key role in alleviating redox-induced cellular injury. The transcriptional activation of NRF2 has been reported to affect both viral replication and associated inflammation pathways. There is complex bidirectional crosstalk between virus replication and the NRF2 pathway because virus replication directly or indirectly regulates NRF2 expression, and NRF2 activation can reversely hamper viral replication and viral spread across cells and tissues. In this review, we discuss the complex role of the NRF2 pathway in the regulation of the pathogenesis of the main respiratory viruses, including coronaviruses, influenza viruses, respiratory syncytial virus (RSV), and rhinoviruses. We also summarize the scientific evidence regarding the effects of the known NRF2 agonists that can be utilized to alter the NRF2 pathway.

## 1. Introduction

Respiratory viruses target the human respiratory system and cause various clinical symptoms in humans, ranging from mild upper respiratory infections to organ failure and life-threatening respiratory diseases [1,2]. The most common respiratory viruses are rhinoviruses, coronaviruses (CoVs), influenza virus, respiratory syncytial virus (RSV), parainfluenza viruses, enteroviruses, adenoviruses, and human metapneumovirus (hMPV) [3]. Each year, nearly 4 million deaths are attributed to lower respiratory tract infections, with Influenza contributing to approximately half a million of these fatalities [4]. Moreover, morbidity and mortality caused by respiratory viruses increased drastically with the emergence of the severe acute respiratory syndrome coronavirus 2 (SARS-CoV-2) and the COVID-19 pandemic. Although several host factors have been found to play crucial role in the pathogenesis of respiratory viral infections, the interaction between respiratory viruses and the host cellular response remains poorly understood. Understanding antiviral host pathways and defining their role in the pathogenesis of respiratory viruses may set the foundation for novel antiviral therapies for viral respiratory diseases.

Viral respiratory infections are commonly associated with increased production of reactive oxygen and nitrogen species (ROS and RNS), leading to oxidative stress [5,6]. Subsequently, increased oxidative stress contributes to reduced host antiviral response, enhanced replication and virus-induced cell and tissue injury apoptosis, ferroptosis, inflammation, causing organ damage [6,7] and the occurrence of clinical symptoms [5,8,9]. During oxidative stress induced by respiratory viruses, the host deploys antioxidant mechanisms to control signaling pathways and reestablish cellular redox balance. Many respiratory viruses, including influenza [10,11,12,13,14,15,16,17], CoVs [18,19,20,21,22,23,24,25,26], RSV [27,28,29,30,31,32,33,34,35,36,37,38,39,40], rhinoviruses [41,42,43,44,45], enteroviruses such as Coxsackievirus B3 (CVB3) [46], EV71 [47,48,49,50], metapneumoviruses [34], and parainfluenza viruses [51,52], have been demonstrated to disrupt the cell redox homeostasis and induce the production of ROS.

However, in response to virus-induced oxidative stress, host cells deploy a strong antioxidant response characterized by the production of proteins (enzymes) and/or small molecules (vitamins C and E) that are mainly mediated by the nuclear factor erythroid 2-related factor (NRF2) to counteract the redox-induced toxicity and restore cellular redox homeostasis [53,54]. In this review, we discuss the intricate role of the NRF2 pathway on the pathogenesis of viral respiratory infections induced by major respiratory viruses, including CoVs, influenza, RSV, and rhinoviruses. Specifically, we focused on the impact of NRF2 activation on the replication of respiratory viruses and summarized the scientific evidence on how certain respiratory viruses dysregulate the NRF2 activation pathway. Obtaining insights into the crosstalk between respiratory viruses and the NRF2 pathway will set the foundation for the use of established NRF2 activators as therapeutics for viral infections.

## 2. The NRF2 Pathway Regulates Cellular Responses to Stress

The production of ROS and activation of an antioxidant response is known to be controlled by the Kelch-like ECH-associated protein 1 (KEAP1)–NRF2 axis. This regulation occurs through intrinsic mechanisms within different cell types of the airway epithelium (e.g., nasal versus bronchial cells). NRF2 belongs to the cap “n” collar (CNC) family of transcription factors and is found in the cytoplasm of non-stressed cells in a combined form with KEAP1. In quiescent cells, an adapter protein, KEAP1, interacts with NRF2 and recruits cullin-3 (CUL3)-containing E3 ubiquitin ligase to form a complex that regulates the ubiquitination of NRF2. Consequently, polyubiquitination of NRF2 leads to NRF2 degradation via the 26S proteasome machinery, which ensures that the NRF2 level and its activity remain low during redox homeostasis [55]. Contrarily, during a viral respiratory infection or other induced oxidative stress, NRF2 escapes repression by KEAP1. The CUL3/KEAP1 complex that targets NRF2 for ubiquitination undergoes a change to a nonfunctional conformation [56,57,58,59]. Thus, upon activation, newly synthesized NRF2 is no longer ubiquitinated/degraded, rapidly accumulates, and translocates to the nucleus where it binds the small maf protein (sMaf) [56,57,58,59]. The NRF2–Maf heterodimer binds to the antioxidant response element (ARE) (or multiple Maf recognition elements (MAREs)). This interaction induces the transcription of a wide variety of antioxidant genes, including HO-1 and genes that are involved in the synthesis and recycling of glutathione (Figure 1). A heme sensor known as BTB and CNC homology 1 (BACH-1) can also bind to the ARE in a KEAP1-independent manner and directly competes with NRF2 for binding to AREs. The interaction of BACH-1 with ARE prevents NRF2 from binding to the ARE, thus repressing HO-1 [60,61,62,63,64,65] (Figure 1). HO-1 catalyzes the degradation of heme into carbon monoxide (CO), Fe^2+^, and biliverdin, and has antiviral properties through multiple pathways (Figure 2). Importantly, NRF2 activation appears to have an inhibitory effect on the interferon response, which is an important component of the innate immune system’s antiviral defense. The balance between NRF2 activation and the interferon response is regulated by intrinsic cell factors, for which the cell composition varies from nasal to bronchial cells, and potentially influences the susceptibility to viral infections. Notably, activation of the NRF2 pathway appears to mediate protection against viral respiratory infections, including SARS-CoV-2, influenza viruses, and several RNA and DNA viruses that induce oxidative stress (Appendix A) [5,8,9,10,11,12,13,14,18,19,20,21,22,23,24,27,28,29,30,31,32,33,34,35,36,41,42,43,44,46,47,48,49,51,52,53,54,65,66,67,68,69,70,71,72,73,74,75,76,77,78,79,80,81,82,83,84,85,86,87,88,89,90,91,92,93]. Overall, the crosstalk between NRF2 and viruses is bidirectional and complex. With regards to the impact of viruses on the NRF2 pathway, respiratory viruses such as influenza [12,94], SARS-CoV-2 [19,20,22,24,95,96], RSV [27,31,32], rhinovirus [43,44], and enteroviruses [47] directly alter NRF2 levels and signaling [5,8,9,10,18,33] (Table 1). With regards to the impact of the NRF2 pathway on viruses, host NRF2 pathways also regulate the replication of several respiratory viruses such as influenza [11,12,13], coronaviruses [19,20,21,22,23], RSV [34,35], rhinovirus [44], enterovirus [47], metapneumovirus [34], and parainfluenza [51,52] (Table 2). Notably, the host NRF2 pathways also regulate viral replication, apoptosis, ferroptosis, and inflammation (Table 2, Table 3, Table 4, Table 5 and Table 6). Besides playing an essential role in cell defense against redox stresses by trans-activating cytoprotective genes encoding antioxidant and detoxifying enzymes, NRF2 contributes to the regulation of the anti-inflammatory response and metabolic reprogramming [97].

## 3. Protective Role of NRF2 in Viral Replication and Pathogenesis

### 3.1. NRF2 Pathway Activation Prevents Viral Entry

The pathogenesis of numerous viral infections appears to entail the NRF2 pathway. First, NRF2 has been implicated in preventing viral entry. The in vitro inhibition of NRF2 is associated with significant enhancement of influenza entry and replication in nasal epithelial cells [11]. Similarly, NRF2 deficiency upregulates the expression of the coronaviruses’ receptor ACE2, enhancing SARS-CoV-2 cell entry and replication [20]. Moreover, it was shown that the lack of the *Nrf2* gene in mice was correlated with enhanced SARS-CoV-2 infection and increased clinical signs of COVID and lung injury when the mice were infected with SARS-CoV-2 [23] (Figure 3). Furthermore, in vitro, the sustained release of ROS allows viruses such as Kaposi’s sarcoma-associated herpesvirus (KSHV) to enter endothelial cells [87]. Importantly, the KEAP1–NRF2 pathway uses NADPH oxidase to control the ROS generated by cellular metabolic functions in the mitochondria and cytosol [122]. Both mitochondrial ROS (mito-ROS) [123] and cytosolic ROS [124] impact endocytosis and endosome formation, which are implicated in the entry of viruses [125,126]. Thus, through its antioxidant properties, NRF2 activation may contribute to reducing viral entry.

### 3.2. NRF2 Pathway Activation Hampers Cytosolic Viral Protein Synthesis

During oxidative stress, NRF2 escapes repression by KEAP1 and new protein synthesis of NRF2 is required for NRF2 activation [56,57,58,59]. Thus, the inhibition of global protein synthesis during acute oxidative stress can paradoxically suppress NRF2 activation [127]. Aberrant redox cellular responses can directly impact protein synthesis [128]. Viruses may seize the translational machinery of their host cells to generate the polypeptides required for viral replication [129]. Rhinoviruses and CoVs are examples of positive-sense single-stranded RNA viruses; hence, their genome serves directly as an mRNA template and is translated in the cytoplasm by the ribosomes of the host, leading to the production of viral polyproteins [130]. In contrast, the genome of negative-strand RNA viruses, such as influenza and RSV, must be converted to their complementary positive-sense RNA which then serves as a template for the synthesis of the new negative-sense viral genomes [131]. Other viruses alter cellular proteins through the NRF2 pathway. Certain viral proteins, such as Hbx in the Hepatitis B virus (HBV), impact the immunoproteasome through the NRF2 pathway to reduce antigen processing and induce a viral escape of the immune system [55,132]. Thus, through its antioxidant properties [122], activating the KEAP1–NRF2 pathway may hinder viral replication by decreasing the virus-induced hijacking of the host protein machinery (Table 2).

### 3.3. NRF2 Activation-Induced Upregulation of HO-1 Inhibits Viral Replication

Upon NRF2 stimulation, heme oxygenase-1 (HO-1) is expressed as an anti-inflammatory enzyme. In response to homeostatic imbalances, HO-1 is upregulated and protects the cell by catalyzing the rate-limiting step of heme degradation and thereby generates carbon monoxide (CO), free iron (Fe^2+^), and biliverdin (BV), which is converted by biliverdin reductase to bilirubin (BR) (Figure 2). Recently published data indicates that HO-1 has antiviral action over a wide range of viruses, including SARS-CoV-2, influenza virus, RSV, and many others [133,134]. NRF2 induces the expression of HO-1, generating Fe2+ that can bind to the divalent metal-binding pocket of the RNA-dependent RNA polymerase (RdRp) of SARS-CoV2 and inhibit its catalytic activity [21] (Figure 2).

Besides the antiviral activity of HO-1 due to the upregulation of type 1 IFN expression [14], HO-1 exerts antiviral activity due to the final products of its degradation. CO, Fe^2+^, BV, and BR can serve directly as antivirals, preventing viral replication through the blockage of the RNA-dependent RNA polymerase of the virus and viral proteases [135]. More specifically, there is evidence that Fe^2+^, can directly bind to the RNA-dependent RNA polymerase of SARS-CoV-2, inhibiting its activity and thus viral replication and progression of infection [19]. The direct binding of iron to the Mg^+2^ binding region of the HCV virus’s RNA polymerase has also been identified to obstruct the virus from replicating [92]. CO has been reported to have a negative impact on enterovirus 71 replication, which can be reversed after the use of compounds with inhibitory activity against CO [48]. Furthermore, bile pigments have been identified to act as inhibitors for the protease of HIV, interfering with the life cycle of the virus [93]. In addition, it has been revealed that bilirubin exhibits an antiviral effect against EV71, decreasing its multiplication and, consequently, infectivity in vitro [49]. Therefore, HO-1 induction, which is mediated by NRF2 activation, can be used as a potential antiviral therapeutic approach against a plethora of different viruses including influenza [14], RSV [36], HCV [88,89,90,91,92], coronaviruses [19], enteroviruses [48,49], and HIV [93] (Table 3).

### 3.4. NRF2 Pathway and Interferon Antiviral Responses during Infection with Respiratory Viruses

Increased HO-1 expression has been revealed to inhibit viral replication during Influenza-A infection by upregulating interferon-stimulated genes (ISGs) [14]. In vitro studies have shown that when NRF2 is overexpressed, there is a reduction in the replication of IV, whereas the knock down of NRF2 leads to enhanced virus entry and replication [11,12]. Influenza strains are thought to activate the NRF2/ARE defense system in vitro and in mice by generating oxidative stress and NRF2 transcriptional activity [12,94]. Similarly, it has been demonstrated that the pharmacological activation of HO-1 by cobalt protoporphyrin (CoPP), a well known heme oxygenase 1 inducer, inhibited the viral replication of RSV in lung cells via the induction of IFN expression [36]. Notably, SARS-CoV-2 interacts with several cellular pathways that crosstalk with the NRF2 pathway. Binding of the viral spike (S) protein to ACE2 leads to virion entry. The viral nucleocapsid is uncoated in the cytoplasm of the host cell (Figure 3) [136]. The virus uses the host machinery to translate viral positive-sense single-stranded RNA (+ssRNA) and cleave the translation product into specific viral proteins. Viral RNA inside the host cell activate the DNA/RNA sensor cGAS, which signals through the adaptor STING [137]. NRF2 represses IFN production by downregulating STING expression. Finally, other respiratory viruses, such as human rhinovirus, stimulate the innate immune sensor RIG-I within airway epithelial cells and activate the antiviral interferon response (and the NRF2-mediated response to oxidative stress) [43]. Thus, the crosstalk between interferons and the NRF2 pathway is important in the pathogenesis of several respiratory viruses.

### 3.5. Cytoplasmic Autophagy Machinery and NRF2 Pathway

Although the NRF2 pathway is a crucial modulator of autophagy [8], there is a complex bidirectional crosstalk between autophagy and the NRF2 pathway [8]. Host defense is conducted by double-stranded RNA-activated protein kinase R (PKR), which phosphorylates eIF2 and inhibits protein translation. This inhibition of protein synthesis is relevant to the inhibition of global protein synthesis during acute oxidative stress that can paradoxically suppress NRF2 activation [127]. PKR also phosphorylates p62, thus activating NRF2 upon the removal of its repressor KEAP1 through autophagy. Besides NRF2, mito-ROS also regulates autophagy [123]. Furthermore, it has been importantly shown that certain viruses can regulate the cell cycle progression and autophagy pathways to enhance their replication [138]. Specifically, once double-membrane vesicles (DMVs) are formed during the autophagy process, the viral replication machinery may employ them as a physical platform, retaining the viral RNA shielded from cleavage and degradation [138]. The way viruses modulate the autophagy pathway is very important for viral pathogenesis, since they can induce variable effects in the cells, promoting their survival or leading to cellular death [139]. Certain viruses, such as Sindbis virus (SINV), herpes viruses, hepatitis viruses, and HIV-1, suppress autophagy to increase their chances of survival [138]. In contrast, to facilitate their replication, some viruses, such as enteroviruses (poliovirus, hand-foot-and-mouth disease virus), elevate autophagy.

Out of the respiratory viruses, CoVs regulate autophagy partly through interactions with NRF2, which thus contributes to viral replication. The crosstalk between NRF2 signaling and endoplasmic reticulum (ER) stress has important implications for proteasomal degradation and autophagy [140]. According to several contentious investigations, CoVs foster autophagy but do not necessitate the entire autophagy mechanism [141,142]. Notably, the NRF2 pathway contributes to the ER stress response in murine and pancreatic cells by boosting proteasome-mediated ER-associated degradation (ERAD) in some mammalian cells, for instance, pancreatic beta cells [143]. During viral infections, such as SARS-CoV-2, the inhibition of protein translation in turn activates the unfolded protein response (UPR). PERK, a crucial Ser/Thr protein kinase in UPR signaling, phosphorylates NRF2, resulting in its stabilization and increased transcriptional activity, and the release of viral particles (Figure 3) [144]. Consistent with the overall evidence that crosstalk between NRF2 signaling, ER stress, and autophagy are critical for the regulation of the pathogenesis of coronaviruses, the non-lipidated LC3-coated EDEMsosome (a vesicle implicated in ERAD) is sequestered through mouse hepatitis virus (MHV) into its replication and transcription subunits [145].

Autophagy is employed for replication by picornaviruses, such as coxsackievirus B3 (CVB3), poliovirus, and foot-and-mouth disease virus (FMDV) [146,147,148,149], through multiple pathways such as galectin 8 [150], the cleavage of p62 through viral proteases [151] or viral proteins [149], and ATG5-dependent autophagosome formation [147]. Similar to flaviviruses, CoVs, and influenza A viruses are the respiratory viruses that assist in the formation of DMVs and autophagosomes to exploit their replication [138]. Human parainfluenza virus type 3 (HPIV3) also regulates autophagy by preventing autophagosome-lysosome fusion [152], which interacts with NRF2 and is involved in viral replication [152]. Viral proteins of HPIV3 also regulate mitophagy [153]. The duration of autophagy activation is also important, since the short-term induction of autophagy has differential downstream cellular effects compared to the sustained induction of autophagy [8]. Overall, depending on the virus and its impact on autophagy and redox cellular responses, activation of the NRF2 pathway can have complex proviral or antiviral effects.

### 3.6. NRF2-Induced Inhibition of Viral Replication Entails Further Biological Mechanisms such as Sirtuins

Sirtuins and NRF2 pair up to boost antioxidant redox signaling and preserve redox equilibrium. Sirtuins are nicotinamide adenine dinucleotide (NAD)-dependent protein deacetylases that regulate several cellular processes such as apoptosis, cellular senescence, and aging [154,155]. SIRT2 specifically deacetylates NRF2 and increases its stability, rescuing NRF2 from degradation [156]. In the absence of SIRT, NRF2 expression is inhibited, resulting in inadequate antioxidant and anti-inflammatory defense for the cell. SIRT1 knockout mice have been demonstrated to hinder NRF2 expression [155]. SIRT1 is the most widely investigated sirtuin in mammals, and its redox modulation is known to be closely related to inflammation resolution and cellular senescence. Along with other sirtuins, SIRT1 was also shown to acquire antiviral activity against a set of DNA and RNA viruses, including herpesviruses, adenovirus, and vesicular stomatitis virus [98,157,158]. Moreover, the HIV-1 Tat protein hinders SIRT1 deacetylase activity [159,160].

Notably, SARS-CoV-2 also interacts with sirtuins, and SIRT1 is inhibited through its interaction with nonstructural SARS-CoV-2 protein NSP14 [95]. The antiviral efficacy of SIRT1 against SARS-CoV-2 is also supported by genetic and pharmacological findings [95]. It is also reported that oxidative stress upon infection with influenza virus leads to the increased acetylation of a redox homeostasis mediator, Glucose-6-phosphate dehydrogenase (G6PD), and it is strictly dependent on the expression of SIRT2. Additionally, in the presence of a SIRT2 pharmacological stimulator, it has been shown that NRF2 expression is rescued, resulting in reduced influenza virus replication [161]. Moreover, Kim et al., using airway epithelial cells, demonstrated that there is crosstalk between the replication efficiency of influenza virus and cellular senescence [162]. When SIRT1 is knocked out in the cells infected with influenza virus, there is an enhanced expression of viral proteins leading to a reduction in the host cell’s viability [162]. Several studies also suggest that during virus-induced airway inflammation SIRT1 regulates neutrophilic abundance related to cytokine CXCL8 [163,164]. The mechanism by which SIRT1 regulates this pathway involves the activation of NF-κB signaling from the NRF2/NF-κB axis, and the subsequent deacetylation of the FOXO 3a protein leading to neutrophilic airway inflammation [163]. RSV infection has also been linked to elevated FOXO3a expression [165]. Therefore, NRF2 activators and/or SIRT activators have been proposed as possible treatment agents for a broad range of viral infections, including respiratory virus infections such as RSV, COVID-19, influenza, adenoviruses, and many other viruses [155,166].

## 4. Respiratory Viruses Leverage or Inhibit NRF2 Activation to Enhance Replication

Overall, the crosstalk between NRF2 and viruses is bidirectional and complex, since respiratory viruses directly alter NRF2 levels and signaling while host NRF2 pathways also regulate viral replication, apoptosis, ferroptosis, and inflammation. Some viruses, however, have established strategies to leverage NRF2 activation and the subsequent antioxidant activity to aid in replication and facilitate pathogenesis [55,167]. Other viruses trigger ROS-independent NRF2 activation, which promotes the pathogenesis of viral infections, tissue injuries, and organ damage [55]. The intricate two-way communication between certain viruses and the NRF2 pathway arises from the viruses’ ability to exert direct or indirect control over NRF2 expression and reversely, from advantages of this controlled activation of NRF2, to exert an influence on the replication and spread of viruses within cells and tissues. External or environmental factors like exposure to cigarette smoke can also trigger ROS-induced NRF2 activation, further impacting the delicate balance between NRF2-mediated antioxidant responses and the interferon response in airway epithelial cells. This suggests that environmental cues can influence the interplay between NRF2-mediated host defense mechanisms and potentially affect the outcome of viral infections in the respiratory tract.

Notably, to further emphasize the complex crosstalk between viruses and NRF2, certain viruses like influenza, hepatitis B virus (HBV), Dengue virus (DENV), and some herpes viruses (HSV1, KHSV), have been demonstrated to upregulate the NRF2 pathway. However, other viruses, such as rhinoviruses, enteroviruses (EV71, CVB3), and hepatitis C virus (HCV), downregulate the NRF2 pathway in infected cells. In addition, there are viruses like SARS-CoV-2 and RSV that, depending on the stage of the infection, can both upregulate and downregulate the NRF2 pathway to their benefit [55] (Figure 3). Respiratory viruses may either leverage NRF2 activation or inhibit this pathway as an immune escape mechanism, which also occurs during respiratory viral infections (Figure 3). This suggests that there are regulated bidirectional mechanisms between viruses and the NRF2 pathway that depend on the type of virus, stage of infection, and the redox context of the infection.

First, it has been demonstrated that SARS-CoV-2 inhibits NRF2 in lung cells, as one of the mechanisms to escape host antiviral immunity [23]. For instance, in COVID-19 patients, enhanced viral replication and severity of the infection were correlated with the dysregulation of NRF2 expression and inhibition of NRF2-dependent antioxidant genes. More specifically, Olagnier et al. demonstrated that the expression of NRF2-dependent genes was hampered in biopsies from COVID-19 patients and this was accompanied by a disrupted redox homeostasis, which resulted in the enhancement of SARS-CoV-2 replication and favored its pathogenesis [22]. Consequently, upregulating NRF2 by any means would prevent SARS-CoV-2 replication and its associated injuries, and improve COVID-19 patient conditions [22] by resolving COVID-19 associated inflammation and cellular redox and protein homeostasis [19]. Thus, the NRF2 pathway has a major role in the pathogenesis of major respiratory virus diseases such as SARS-CoV-2 infection (Figure 4).

Second, upon RSV infection the redox homeostasis is disrupted with the production of ROS. As for SARS-CoV-2 infection, it has been revealed that RSV would degrade NRF2 as an immune escape mechanism, which would consequently inhibit the expression of antioxidant enzymes. Specifically, an RSV infection induces the deacetylation, ubiquitination, and degradation of NRF2 through a proteasome-dependent pathway, resulting in very low levels of NRF2, viral replication, and an exacerbation of the infection [30,168].

Third, a recent study has demonstrated that to escape from a host immune response and to sustain viral replication, Influenza A virus (IAV) indirectly downregulates the production of the NRF2-associated antioxidant response. Especially, IAV is an intracellular parasite that uses host proteins, including proteasome subunit alpha type 2 (PSMA2), to complete its replication cycle and suppress immune responses. In the absence of PSMA2, the viral replication is almost inexistent. Thus, IAV uses PSMA2 to inhibit the NRF2 signaling pathway and escape an antiviral response [169]. This suggests that the activation of NRF2 and/or inhibition of PSMA2 can serve to mitigate viral infections and reduce the associated burden.

Unlike coronaviruses, RSV, and influenza virus, rhinovirus infection does not systematically up- or downregulate NRF2. In their study, Andrew et al. [170] demonstrated that, depending on the infected cells, rhinovirus infection can upregulate or downregulate the NRF2 pathway. More precisely, in monocytes but not in epithelial cells, rhinovirus infection upregulates NRF2, although increased inflammation was observed in both cells. Therefore, in such an infection, before thinking of developing an NRF2 agonist, it is mandatory to perform in-depth studies to understand what drives each phenomenon differently.

## 5. NRF2 Pathway Contribution in Viral Pathogenesis in Cell and Tissue Damage

The NRF2 pathway is involved not only in viral replication pathways (virus entry, protein machinery, autophagy, HO-1), but also in associated virus-induced pathways that regulate cell and tissue damage such as inflammation, apoptosis, autophagy, and ferroptosis. Inflammation is a complex and highly regulated immune response that occurs in response to virus-induced alterations in tissue homeostasis. NF-ĸB is activated by virus-induced oxidative stress leading to the exacerbation of pro-inflammatory cytokines in the airway [171]. Importantly, the activation of the NRF2/ARE system alters this cycle. The NRF2 pathway is linked to a considerable reduction in the LPS-induced transcriptional upregulation of pro-inflammatory cytokines (i.e., IL-6 and IL-1) [102]. NRF2 depletion promotes the NF-kB pathway to activate and elevate cytokine production [102]. Generally, during a viral infection the inflammatory reactions target the clearance of the infection and the subsequent resolution of the virus-induced inflammation. Influenza [15,16,17], coronaviruses [23,25,26], RSV [37,38,39,40,102], human metapneumovirus (hMPV) [34], enteroviruses [47,50], and rhinoviruses [16,45] are among the most common viruses known to also exacerbate airway inflammation [172]. The complex crosstalk between respiratory viruses, the NRF2 pathway, and inflammatory responses is summarized in Table 4.

Respiratory viruses such as Influenza A virus can induce redox damage, leading to a redox imbalance in the host cell and subsequently to associated cell and tissue damage [173,174]. In vitro and in vivo evidence suggests that the NRF2 pathway is a critical factor for both influenza replication reduction (Table 3) and the host immune response against influenza. NRF2 has also been demonstrated to play a significant role in inflammation and the cytokine storm induced by coronavirus infection, including SARS-CoV-2 infection [23,25,26]. The newly emerged coronavirus and its variants, COVID-19, can lead to a cytokine storm, extreme immune responses, widespread inflammation, and oxidative stress, contributing to organ damage. Oxidative stress plays a significant role in the pathophysiology of COVID-19 and is linked to the development of organ damage in severe cases of the disease, leading to acute respiratory distress syndrome (ARDS), a life-threatening condition [175]. Overall, the activation of NRF2 can exploit a broad anti-inflammatory effect against a variety of viral pathogens (Table 4). For this reason, NRF2 and its modulators can serve as promising targets for the suppression of viral infection-associated inflammatory reactions and their implications.

### 5.1. Virus-Induced Apoptosis and the NRF2 Pathway

NRF2 appears to protect against a variety of viral infections and their associated host responses. NRF2 has been identified to have an antiapoptotic function. Apoptosis, a form of programmed cellular death, is a key cellular response to viral infection [103]. The induction of apoptosis from virus-infected cells is a defense mechanism to halt the generation, release, and further spread of viral particles. Several viruses have devised various techniques to manipulate their host’s cellular death for their own gain. Some viruses can prevent the premature death of infected cells and therefore establish a persistent latent infection, while others enhance the duration of their lytic infection and as a result increase the production and release of viral progeny [103,104]. Experimental studies have shown that autophagy plays a very crucial role in the replication of several respiratory viruses like RSV [107]. The complex crosstalk between the NRF2 pathway, apoptosis, and respiratory viruses such as influenza [15,16,17], coronavirus [23,25,26], RSV [37,38,39,40,102], metapneumovirus [34], enterovirus [47,50], and rhinovirus [16,45], is summarized in Table 5.

However, the mechanisms that mediate the interaction between NRF2 and known apoptotic pathways remain unclear. It has been found that in cancer cell lines, NRF2 elevates Bcl-2 expression, attenuates proapoptotic Bax protein and caspases 3/7, and rescues cells from etoposide/radiation-induced apoptosis [176]. Moreover, p53 can trigger apoptosis through a variety of signaling pathways including reduced NRF2 activity [177]. Additionally, NRF2 influences the function of Glutathione S-transferase Pi (GSTP1), which has the ability to impede the proapoptotic Jun N-terminal kinases (JNKs) from performing their normal functions [178]. Likewise, NRF2 upregulates HO-1, which in turn can suppress TNF-alpha, a potent mediator of cell death [179]. Taken together, this implies an established link between NRF2 and virus-induced apoptosis, which is predominantly mediated by the redox status of the cell microenvironment.

### 5.2. Ferroptosis and the NRF2 Pathway

An alteration in a cell’s redox equilibrium, primarily induced by the buildup of ROS, triggers the iron-dependent process of cellular death known as ferroptosis [180,181]. The iron concentration in a cell has a very critical role in the ferroptosis pathway, since elevated levels of iron lead to Fenton reactions and subsequently ferroptosis [121]. It has been shown that multiple viruses, including both non-respiratory [121] and respiratory viruses, such as influenza [9,108,109,110,111,112], coronaviruses [113,114,115,116,117], RSV [118], and enteroviruses [108,119,120], can induce ferroptosis during an infection by regulating iron uptake and metabolism (Table 6). Viral proteins like ORF3a from SARS-CoV-2 recruit KEAP-1, which inhibits NRF2 activity, facilitating ferroptosis through the build-up ROS and the downregulation of genes like HO-1 and NQO1 [96]. Many essential viral activities, such as viral replication and the expression of viral proteins, also promote ferroptosis by interfering with the metabolism of the host. Consequently, the Krebs cycle, electron transport chain activity, and glutaminolysis are enhanced [121].

NRF2 has a regulatory effect on ferroptosis by modulating the expression of multiple key genes of the ferroptosis pathway. More specifically, NRF2 can directly upregulate glutathione (GSH) and glutathione peroxidase 4 (GPX4) synthesis, which are very important inhibitors of ferroptosis [182]. In addition, numerous studies indicate that NRF2 modulates the expression of HO-1, which subsequently affects the iron content in a cell [183,184,185]. For these reasons, NRF2 can be considered as a potential target to suppress ferroptosis.

## 6. Therapeutic Potential of NRF2 Activation

NRF2 inducers are being used as a therapeutic approach for a variety of diseases that cause chronic inflammation, including multiple sclerosis and lupus erythematosus [186,187]. NRF2 can also be a promising therapeutic target for many viral diseases associated with oxidative stress and inflammation. Given the severity of the inflammatory component during the viral infections described above and the role of NRF2 activation in the progression of disease, it seems that the NRF2 pathway can serve as a broad antiviral and anti-inflammatory target. Various studies have shown the therapeutic benefits of NRF2 activators in infections with SARS-CoV-2 influenza, parainfluenza, and RSV (Table 7) [11,37,51,168,188,189,190,191,192,193].

NRF2 agonists such as 4-octyl itaconate (4-OI), the cell-permeable form of the metabolite itaconate, have been tested in vitro against SARS-CoV-2 infection. 4-OI serves to alkylate cysteine residues on KEAP-1, thereby freeing NRF2 to promote the expression of oxidative damage-reduction genes (HO-1 and GSH), and was shown to downregulate STING and IFN pathways [32,195]. On the other hand, DMF is an NRF2 agonist and FDA-approved drug for multiple sclerosis, and was also shown to be effective against a wide range of viruses including SARS-CoV-2 [22]. Both DMF and 4-OI were found to abolish the proinflammatory state caused by SARS-CoV-2 in Calu-3 cells, possibly through the increased activity of HO-1 [22].

Another naturally occurring compound with antiviral activity surrounding NRF2 activation is Curcumin, which is found in turmeric. Curcumin has been shown to reduce influenza, parainfluenza, and RSV replication, lowering the oxidative stress caused by viral infection likely through HO-1 activation [51,191,192]. Isothiocyanate sulforaphane (SFN) naturally occurs in broccoli and is another compound that has been shown to have antiviral properties through NRF2 activation and the induction of NQO1 activity [189]. SFN has been shown to have antiviral properties against RSV and is ineffective in NRF2 knockout mice [37]. SFN has also been shown to lower IL-6 values and the viral load of influenza A in smokers who were given an attenuated form of the virus [190]. Epigallocatechin-3-0-gallate (EGCG) is an NRF2 activator and has proven anti-influenza activity by blocking viral entry and a subsequent reduction in viral replication [16]. Moreover, carbocisteine, another activator of NRF2, led to a diminished expression of the nucleoprotein of the virus and thus viral replication [193]. Additionally, butylated hydroxyanisole (BHA) and the related metabolite tert-butyl hydroquinone (tBHQ) have been shown to activate the expression of antioxidant genes such as HO-1, NQO1, and NRF2 itself [168].

However, most of the experimental evidence that supports the use of NRF2 agonists as treatment for viral infections is based on in vitro experimental studies that were not validated in vivo. Notably, the balance in regulation of the NRF2 pathway was not considered in these in vitro experimental studies that were either based on the complete inhibition of NRF2 or the use of supraphysiological concentrations of NRF2 agonists that may not be translated in humans. For example, even though NRF2 can have a positive impact against a plethora of diseases, it is important to maintain its levels in balance. It has been shown that the overactivation of NRF2 and aberrant levels of NRF2 can lead to the uncontrollable proliferation and growth of cancer cells and prevent their apoptosis [167,177,196]. In vivo experimental studies in mice [194] and clinical trials in humans with the NRF2 agonist sulforaphane [197] have also provided valuable insights into the use of Nrf2 agonists in vivo [194]. Additionally, NRF2 activation may help prevent ferroptosis by promoting autophagy and targeting genes involved in glutathione (GSH) synthesis and metabolism [198]. Thus, depending on the infection, the associated disease, and the context, the overactivation of NRF2 may have detrimental effects in humans. Thus, this dual effect of NRF2 highlights that the usage of NRF2 agonists as a therapeutic approach needs to be studied carefully in order to prevent excessive activation.

## 7. Conclusions

NRF2 has a key role in a broad range of infections caused by respiratory viruses such as influenza, coronaviruses, and RSV. NRF2 activation has a multifactorial effect in viral infections, including the direct inhibition of viral replication and its implications for the host, including virus-induced apoptosis, ferroptosis, and inflammation. Taking into consideration the significance of NRF2 in the antiviral response and the development of disease, regulation of the NRF2 pathway has the therapeutic potential to control virus-induced pathologies. However, most of the experimental evidence that supports the use of NRF2 agonists as a treatment for viral infections is based on in vitro experimental studies that were not validated in vivo. Notably, activation of the NRF2 pathway can be a double-edged sword, since the overactivation of NRF2 can lead to an uncontrollable proliferation and growth of cancer. Further studies in animals and humans are needed to study the use of NRF2 agonists in the presence of a physiologically meaningful activation as a potentially useful therapeutic strategy for infections from respiratory viruses.

## Figures and Tables

**Figure 1 pathogens-13-00039-f001:**
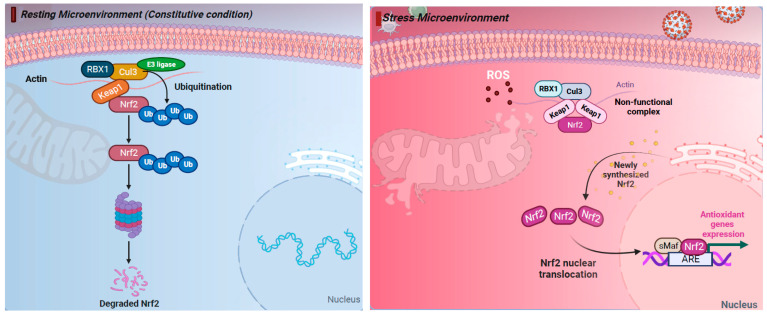
The nuclear factor erythroid 2-related factor 2 (NRF2) pathway regulates cellular responses to stress. (Left) Under resting (constitutive) condition, in the cytoplasm NRF2 is anchored with Kelch-like ECH-associated protein 1 (KEAP1). NRF2 binds KEAP1 and becomes ubiquinated, leading to degradation by the 26S proteasome. (Right) Under oxidative stress response, NRF2 escapes repression by KEAP1. The CUL3/KEAP1 complex that targets NRF2 for ubiquitination undergoes a change to a nonfunctional conformation. Thus, newly synthesized NRF2 is no longer ubiquitinated/degraded, rapidly accumulates, and translocates to the nucleus where it binds the small maf protein (sMaf) and antioxidant response element (ARE). Activation of ARE increases the expression of the antioxidant genes heme oxygenase 1 (HO-1), quinone oxidoreductase (NQO1), and glutathione (GSH), which blocks the progression of oxidative stress (OS). Thus, activation of the NRF2 pathway has cytoprotective effects and plays a key role in maintaining redox balance. Figure generated with Biorender (https://biorender.com/, accessed on 10 November 2023).

**Figure 2 pathogens-13-00039-f002:**
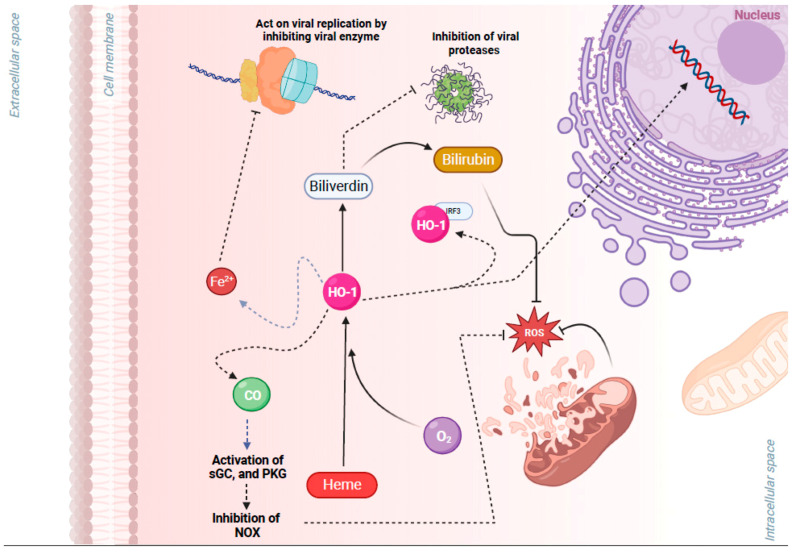
Cytoprotective effects of heme oxygenase 1 (HO-1), a key gene of the nuclear factor erythroid 2-related factor 2 (NRF2) pathway, in viral infection. HO-1 is a metabolic enzyme that utilizes oxygen (O_2_), heme, and NADPH to catalyze the degradation of heme into carbon monoxide (CO), Fe^2+^, and biliverdin. HO-1 has antiviral properties through multiple pathways: (1) Free Fe^2+^ may act on viral replication by binding to the highly conserved divalent metal-binding pocket of the viral RNA and inhibiting enzymes that mediate viral replication. (2) Biliverdin may inhibit viral proteases. (3) Heterodimerization of HO-1 with interferon regulatory factor 3 (IRF3) facilitates the phosphorylation and nuclear translocation of IRF3 and the induction of type I interferon (IFN) gene expression that has antiviral properties. (4) CO activates protein kinase G (PKG), which inhibits NAPDH oxidases (NOX), preventing an increase in reactive oxygen species (ROS) and associated damage. (5) Biliverdin also has antioxidant properties, and it is converted by NRF2/ARE-regulated gene biliverdin reductase to the potent antioxidant bilirubin. Figure generated with Biorender (https://biorender.com/, accessed on 10 November 2023).

**Figure 3 pathogens-13-00039-f003:**
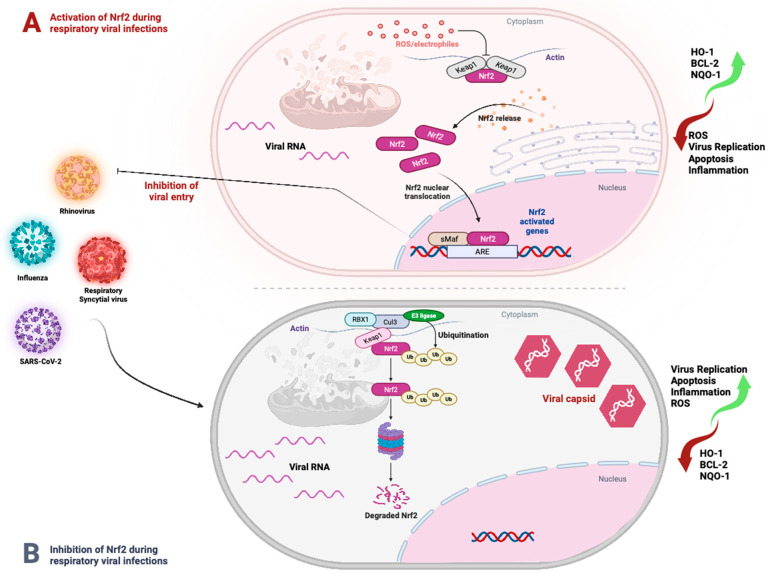
Bidirectional crosstalk between replication of respiratory viruses and NRF2. Depending on the phase of the infection (early versus late), the type of respiratory virus and the context of the redox responses, respiratory viruses can either activate (**A**) or inhibit (**B**) the NRF2 pathway and associated downstream genes such as HO-1, NQO1, and BCL2, which regulate ROS, inflammation, apoptosis, viral entry, and replication. Figure generated with Biorender (https://biorender.com/, accessed on 10 November 2023).

**Figure 4 pathogens-13-00039-f004:**
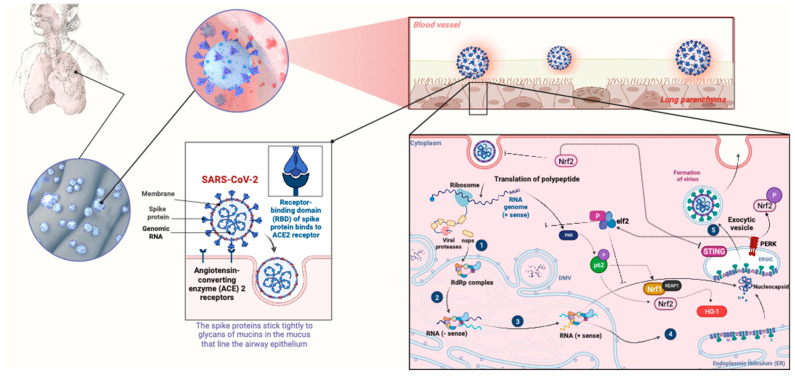
Crosstalk of SARS-CoV2 with the NRF2 pathway. The viral spike (S) protein binds to the receptor angiotensin-converting enzyme 2 (ACE2), leading to virion entry. The viral nucleocapsid is uncoated in the cytoplasm of the host cell and the viral positive-sense single-stranded RNA (+ssRNA) is translated and then cleaved into specific viral proteins. Autoproteolysis and co-translational cleavage of polypeptide to generate nsps is shown as (**1**)). (- Sense) subgenomic transcription and RNA replication is shown as (**2**). (+ Sense) subgenomic transcription and RNA replication is shown as (**3**). Translation of subgenomic mRNA into structural and accessory proteins is shown as (**4**). Viral RNA inside the host cell activates the DNA/RNA sensor cGAS, which signals through the adaptor STING. Host defense is conducted by double-stranded RNA-activated protein kinase R (PKR), which phosphorylates eIF2 and inhibits protein translation. PKR also phosphorylates p62, thus activating NRF2 upon removal of its repressor KEAP1 through autophagy. Inhibition of protein translation in turn activates the unfolded protein response (UPR). PERK, a crucial Ser/Thr protein kinase in UPR signaling, phosphorylates NRF2, resulting in its stabilization and increased transcriptional activity. Activation of the NRF2 pathway (i) represses IFN production by downregulating STING expression; and (ii) induces the expression of HO-1, generating Fe^2+^ that can bind to the divalent metal-binding pocket of the RNA-dependent RNA polymerase (RdRp) of SARS-CoV2 and inhibit its catalytic activity. Thus, the NRF2 pathway is involved in multiple pathways involved in SARS-CoV-2 pathogenesis. Abbreviations: ACE2, angiotensin-converting enzyme 2; eIF2, eukaryotic initiation factor 2; ER, endoplasmic reticulum; ERGIC, ER–Golgi intermediate compartment; HO-1, heme oxygenase 1; IFN, interferon; KEAP1, Kelch-like ECH-associated protein 1; NRF2, nuclear factor erythroid 2 p45-related factor 2; PERK, PKR-like endoplasmic reticulum kinase; P, phosphorylation; PKR, protein kinase R; STING, stimulator of interferon genes. Figure generated with Biorender (https://biorender.com/, accessed on 10 November 2023).

**Table 1 pathogens-13-00039-t001:** Impact of respiratory viruses on the NRF2 pathway.

Viruses	Mechanism of NRF2 Activation	Reference
Several (respiratory) viruses	∙ ↑ ROS and mito-ROS ↑ ARE elements and phosphorylation of p62.	[5,8,9]
Influenza	∙ IV strains are thought to activate the NRF2/ARE defense pathway in vitro and in mice by inducing oxidative stress and nuclear translocation and transcriptional activity of NRF2 because transcription of the NRF2 target gene HO-1 was shown to be augmented.	[12,94]
SARS-CoV-2	∙ SARS-CoV-2 infection ↓ levels of NRF2 in epithelial cells in vitro [19,20,22]. ∙ NRF2 was ↓ in RNA Seq analysis of lung biopsies of COVID-19 patients [22]. ∙ NRF2 deficiency ↑ ACE2, enhancing viral entry and as a result viral replication [19,20]. ∙ The nonstructural SARS-CoV-2 NSP14 viral protein inhibits NRF2 through ↓ of SIRT1 [95]. ∙ The SARS-CoV-2 ORF3a viral protein recruits KEAP-1 which inhibits NRF2 activity, thereby facilitating ferroptosis through the built-up ROS and the downregulation of genes like HO-1 and NQO1 [96]. ∙ The SARS-CoV-2 ORF3a viral protein binds to human HO-1 protein in vitro [24].	[19,20,22,24,95,96]
RSV	∙ RSV deregulates the NRF2 expression and its activity along with the upregulation of downstream ARE-responsive genes [98]. ∙ RSV ↓ mRNA levels of NRF2 in airway epithelial cells [27]. ∙ RSV ↑ NRF2 deacetylation, ubiquitination, and degradation through a proteasome-dependent pathway in a SUMO-specific E3 ubiquitin ligase RNF4-dependent manner. ∙ Another possible mechanism of RSV-associated NRF2 activation is the activation of [31], which activates the NRF2 pathway through direct alkylation of the NRF2 partner—KEAP1 [32].	[27,31,32,98]
Rhinovirus	∙ Rhinovirus RNA stimulates the innate immune sensor RIG-I within airway epithelial cells and activates the antiviral interferon response (greater activation in nasal cells than in bronchial cells) and the NRF2-mediated response to oxidative stress (greater activation in bronchial cells compared to nasal cells) [43]. ∙ However, the inhibitory effects were reversed in cells pretreated with the antioxidant, N-acetyl cysteine. Moreover, the secretion of anti-viral interferons ↑ in cells treated with the NRF2 agonist sulforaphane but ↓ in cells where NRF2 was silenced [44].	[43,44]
Enterovirus 71 (EV71)	∙ EV71 ↑ KEAP1 and ↓ NRF2 [47].	[47]
RSV, influenza, coronaviruses, HCV	∙ ↑ phosphorylation of the redox-sensitive PKC ↑ NRF2 dissociation from KEAP1.	[10,18,33]

Abbreviations: ACE2: angiotensin-converting enzyme, ARE: antioxidant response element, COX-2: cyclooxygenase 2, EBV: Epstein Barr Virus, ER: endoplasmic reticulum, EV71: Enterovirus 71, GSK3: glycogen synthase kinase 3, HBV: Hepatitis B virus, HCV: Hepatitis C virus, HO-1: Heme Oxygenase 1, IRG1: immune-responsive gene 1, IV: influenza virus, mito-ROS: mitochondrial ROS, KEAP1: Kelch-like ECH-associated protein 1, KSHV: Kaposi sarcoma-associated herpesvirus, NQO1: NAD(P)H quinone oxidoreductase, NRF2: nuclear factor erythroid 2-related factor, NSP14: nonstructural protein 14, ORF3a: 3a open reading frame 3a, PKC: protein kinase C, PI3K: phosphatidylinositol 3-kinase (PI3K), PGE2: Prostaglandin E2. RNF4: RING finger protein 4, ROS: reactive oxygen species, RSV: respiratory syncytial virus, SIRT1: sirtuin 1, SFTS: severe fever with thrombocytopenia syndrome, sMafs: small Maf proteins, SUMO: small ubiquitin-like modifiers.

**Table 2 pathogens-13-00039-t002:** Impact of NRF2 on the replication of respiratory viruses.

Virus	Role of NRF2 in Viral Replication	Reference
Influenza virus	∙ Activation of NRF2 leads to ↓ viral replication through interferon host responses ∙ Downregulation of NRF2 results in major ↑ of viral entry and, subsequently, viral replication. ∙ Inhibition of viral replication, growth, and protein expression after activation of NRF2. Influenza also regulates autophagy, which interacts with NRF2 and is involved in influenza replication [99].	[11,12,13]
Coronaviruses	∙ NRF2 deficiency upregulates ACE2, ↑ viral entry, and, as a result, viral replication. ∙ NRF2 induced production of HO-1 and generated Fe^+2^, which binds to the RNA- dependent RNA polymerase of SARS-CoV-2, inhibiting its activity and thus viral replication. ∙ NRF2 agonists like 4-OI and DMF inhibit SARS-CoV-2 replication. ∙ Vero cells infected with SARS-CoV-2 and transfected with siRNA to silence KEAP-1, thereby activating NRF2, had a decreased viral load. ∙ Absence of NRF2 in knockout mice ↑ the severity of SARS-CoV-2 infection and viral replication.	[19,20,21,22,23]
RSV	∙ NRF2 knockout mice showed significantly ↑ viral titers in the lungs. ∙ Treatment of the NRF2 agonist sulforaphane on NRF2−/− and NRF2+/+ mice before RSV infection ↓ virus replication, but this significant effect was not observed in NRF2−/− mice [37]. ∙ Compared to wild-type mice, RSV-infected NRF2 KO had ↓ antioxidant enzymes and enzymes in the airway, which modulated the endogenous hydrogen sulfide (H2S) pathway that has a significant antiviral function [34]. ∙ Inducers of the NRF2-ARE pathway, such as BHA treatment, ↑ the viral clearance in murine lungs [35].	[34,35,37]
Rhinovirus	∙ Silence of NRF2 in cells led to a ↓ in the secretion of antiviral interferons and higher viral titers.	[44]
Enterovirus 71 (EV71)	∙ Silencing of NRF2 is beneficial for viral replication. ∙ Activating NRF2 through downregulation of KEAP1 led to ↓ viral replication in RD cells.	[47]
Metapneumovirus	∙ ↓ of NRF2-dependent genes ↑ viral replication and clinical disease upon hMPV infection.	[34]
Parainfluenza viruses	∙ Cotreatment or post infection treatment with curcumin, ↓ the expression of HN viral protein, indicating that curcumin may ↓ viral entry affecting viral replication and subsequently different steps in viral replication ∙ Curcumin, an NRF2 activator, led to ↓ of F-actin, ↓ the formation of viral IBs, and ↓ viral replication. ∙ Curcumin ↓ HPIV3 replication by ↓ the endogenous PI4KB level in the cells, and ↓ the colocalization of PI4KB and IBs, affecting IB formation.	[51,52]

Abbreviations: ACE2: angiotensin-converting enzyme, ARE: antioxidant response element, BHA: butylated hydroxyanisole, DMF: Dimethyl fumarate, EV-71: Enterovirus 71, hMPV: Human Metapneumovirus, HO-1: Heme Oxygenase 1, HPIV3: Human Parainfluenza virus 3, IBs: inclusion bodies, KEAP1: Kelch-like ECH-associated protein 1, NRF2: nuclear factor erythroid 2-related factor, 4-OI: 4-octyl itaconate, PI4KB: 1-phosphatidylinositol 4-kinase beta, RD: Rhabdomyosarcoma, RSV: respiratory syncytial virus, SARS-CoV-2: severe acute respiratory syndrome-coronavirus 2, siRNA: Small interfering RNA.

**Table 3 pathogens-13-00039-t003:** Antiviral activity of Heme Oxygenase 1 (HO-1), a key downstream gene of the NRF2 pathway.

Viruses/NRF2	HO-1 Antiviral Activity	Reference
Influenza	∙ ↑ expression of HO-1 leads to ↓ viral replication during infection from Influenza A, through the upregulation of IFN- α/β and ISGs.	[14]
RSV	∙ Harmacological activation of HO-1 by CoPP ↓ viral replication of RSV in lung cells of infected mice via induction of IFN-α/β expression. ∙ In vitro data suggest that ↑ of HO-1 can moderate the susceptibility of cells to hRSV infection [36].	[36]
HCV	∙ Type 1 IFN-dependent anti-HCV activity due to ↑ levels of HO-1 resulting from the usage of HO-1 agonists/inducers. ∙ Iron stopped viral replication of HCV by direct binding to the Mg+2 binding pocket of the RNA polymerase of the virus.	[88,89,90,91,92]
Coronaviruses	∙ Fe+2 binds to the RNA-dependent RNA polymerase of SARS-CoV-2 inhibiting its activity and ↓ viral replication. ORF3a protein binds to human HMOX1 protein in vitro [24].	[19]
EV71	∙ The overexpression of HO-1 ↓ NADPH oxidase/ROS production that is induced by enterovirus 71 and hence ↓ viral replication. This effect was abolished if cells were pretreated with zinc–protoporphyrin IX, an HO-1 activity inhibitor. ∙ Bilirubin has also been found to exert antiviral activity against EV71 reducing its replication and as a result infectivity in vitro.	[48,49]
HIV	∙ BV and BR have been identified to act as inhibitors for the protease of HIV, interfering with the life cycle of the virus.	[93]

Abbreviations: CoPP: cobalt protoporphyrin, EV-71: Enterovirus 71, HCV: Hepatitis C virus, HO-1: Heme Oxygenase 1, ISGs: interferon stimulated genes. NRF2: nuclear factor erythroid 2-related factor, SARS-CoV-2: severe acute respiratory syndrome coronavirus 2, RSV: respiratory syncytial virus.

**Table 4 pathogens-13-00039-t004:** Respiratory viruses and the role of NRF2 role in inflammation.

Virus	Role of NRF2 in Inflammation	References
Influenza virus	∙ Inactivation of NF-κΒ transcription factor. ∙ ↓ of NF-κΒ-mediated inflammation and associated lung permeability damage, mucus hypersecretion, lung permeability damage, as well as mucus hypersecretion, through reduced NF-κΒ-mediated inflammation and associated proinflammatory cytokines [15]. ∙ Induces anti-inflammatory effects in vivo through the HO-1 pathway [100,101]. ∙ NLRP3 activation form PB1-F2 influenza A protein. ∙ K+ efflux and ROS dependent activation of NLRP3 inflammasome ∙ Impacts function of alveolar macrophages (AMϕ) that are important essential for preventing respiratory failure and mortality after infection from influenza virus in mice [17]. ∙ Attenuates virus-induced inflammation through increased GSH levels and IL-8 secretion in ATI-like cells (alveolar epithelial cells) in vitro [12].	[15,16,17]
Coronavirus	∙ NRF2 is directly able to inhibit IL6, IL-1B, a key hallmark of the cytokine storm in SARS-CoV-2 infection. ∙ Absence of NRF2 in knockout mice ↑ the severity of SARS-CoV-2 infection, pulmonary inflammation. ∙ In humans, SNPs in the *Nrf2* gene promoter region can determine susceptibility to respiratory failure with COPD, indicating the importance of NRF2 in pulmonary inflammation. ∙ Cytokine storm due to T cell depletion and widespread pulmonary inflammation. ∙ Contradictory effect on proinflammatory nature of factors like NF-kB.	[23,25,26]
RSV	∙ Severe inflammation in NRF2−/− mice compared to NRF2+/+ mice. ∙ RSV-infected NRF2 KO mice are reported to have a significant ↑ in airway neutrophilia and inflammatory cytokines. ∙ ↓ lung inflammation when pretreated with sulforaphane. ∙ ↓ ROS- and K+ efflux-dependent activation of NLRP3 inflammasome. ∙ SH viroporin activates NLRP3 inflammasome. ∙ Impacts function of alveolar macrophages (AMϕ), which are important to attenuate virus-induced inflammation.	[37,38,39,40,102]
Metapneumovirus	∙ NRF2 KO mice infected with hMPV had ↓ expression of antioxidant enzymes (AOE) and ↑ viral-mediated oxidative stress and airway damage compared to NRF2+/+ mice.	[34]
Enterovirus 71	∙ By silencing KEAP1, the induced ROS, apoptosis, and inflammation was ↓ in the EV71 infected cells. However, when both KEAP1 and NRF2 were silenced in Vero and RD cells, these effects were restored. ∙ Inflammation-promoting cytokines and chemokines influence the severity of the EV71 infection.	[47,50]
Rhinovirus	∙ 2B viroporin activates NLRP3 inflammasome	[16,45]

Abbreviations: EV-71: Enterovirus 71, KEAP1: Kelch-like ECH-associated protein 1, hMPV: human metapneumovirus, NRF2: nuclear factor erythroid 2-related factor, NRF2-KO: NRF2 knocked out; NF-κB: nuclear factor kappa B, ROS: reactive oxygen species, RSV: respiratory syncytial virus.

**Table 5 pathogens-13-00039-t005:** Impact of viruses and NRF2 on the apoptosis pathway.

Viruses/NRF2	Impact on Apoptosis	Reference
Adenoviruses	∙ Complex effects. ∙ ↑ apoptosis: ↑ sensitivity to TNFa that induces apoptosis, ↑ PP2A, and ↑ p53. ∙ ↓ apoptosis through several mechanisms: interacts with FADD, ↓ CD95-mediated apoptosis, ↓ phospholipase A2, ↓ Fas, ↓ p53, and ↓ pro-apoptotic proteins of the Bcl-2 family, such as Bax, Bak, BNIP3, and Bnip3L. ∙ ↓ apoptosis of the host cell in order to ↑ efficiently and the capacity of the virus to ‘hijack’ host cell apoptotic machinery.	[103,104,105,106]
RSV	∙ ↑ interferons and caspase 1. ∙ Experimental studies have shown that autophagy plays a very crucial role in RSV replication [107].	[105]
Influenza	∙ ↑ Fas expression. ∙ ↓ PKR and apoptosis. ∙ Apoptosis plays a role in viral release.	[105,106]
Rhinovirus, enteroviruses	∙ ↑ apoptosis through unknown mechanism.	[105]
Coronaviruses	∙ ↑ apoptosis through ORF proteins and unknown mechanisms.	[105]

Abbreviations: FADD: Fas-associated death domain protein, GPX4: Glutathione Peroxidase 4, HO-1: Heme Oxygenase 1, NRF2: nuclear factor erythroid 2-related factor, ORF: Open Reading Frame, PKR: protein kinase R, p53: tumor protein 53, PP2A: Protein Phosphatase 2A, ROS: reactive oxygen species, RSV: respiratory syncytial virus, TNFa: tumor necrosis factor a.

**Table 6 pathogens-13-00039-t006:** Impact of viruses and NRF2 on the ferroptosis pathway.

Viruses/NRF2	Impact on Ferroptosis	Reference
NRF2	∙ NRF2 ↓ ROS and ↑ antioxidant responses, and ↑ GPX4-induced ↓ of ferroptosis. ∙ NRF2 ↑ Heme Oxygenase 1 (HO-1) that ↓ ferroptosis. ∙ NRF2 ↑ antioxidant enzymes.	[9]
Influenza	∙ Iron ↓ viral genome amplification and viral replication. ∙ Influenza ↓ cellular GSH and/or affects GPX4 activity. ∙ Neuraminidase of Influenza A binds lysosome-associated membrane proteins and ↑ lysosome rupture.	[9,108,109,110,111,112]
SARS-CoV-2, SARS-CoV, other coronaviruses	∙ SARS-CoV-2 Potentially causes cellular iron overload and iron scavenging. ∙ SARS-CoV-2 ↑serum ferritin concentration. ∙ CoVs↓ cellular GSH and/or affect GPX4 activity. ∙ SARS-CoV ORF-3a viral protein ↑ lysosomal damage and dysfunction.	[113,114,115,116,117]
RSV	∙ ↑ the expression of 12/15-LOX and mitochondrial iron content.	[118]
EV-71, CB3	∙ Iron ↓ viral genome amplification and viral replication of EV-71. ∙ CB3 ↑the expression NRAMP (DMT) and ↑cellular iron uptake.	[108,119,120]
Non-respiratory viruses: HBV, HCV, WNV, Dengue virus, HSV, KSHV	∙ HBV, HCV: ↑ serum and cellular iron uptake and ↓ hepcidin expression, ↑ serum ferritin concentration, and uses TfR1 as a cellular receptor. ∙ HIV ↓ serum iron, ↑ the expression of hepcidin, ↑cellular iron via hepcidin mediated degradation of ferroportin, ↓ cellular GSH and/or affects GPX4 activity, and upregulates the expression of system xc-. ∙ WNV ↑ the expression NRAMP (DMT) and ↑cellular iron uptake. ∙ Dengue virus ↓ cellular GSH and/or affects GPX4 activity. ∙ HSV ↓ cellular GSH and/or affects GPX4 activity. ∙ JEV ↓ cellular GSH and/or affects GPX4 activity, produces lipid peroxide free radicals, and ↑ the expression of system xc-. ∙ KSHV ↓ cellular GSH and/or affects GPX4 activity. ∙ Zika virus ↓ cellular GSH and/or affects GPX4 activity. ∙ Other viruses (e.g., hemorrhagic viruses) use NRAMP or TfR1 as a cellular receptor.	[121]

Abbreviations: CB3: Coxsackievirus B3, EV-71: Enterovirus 71, GSH: Glutathione, GPX4: Glutathione Peroxidase 4, HBV: Hepatitis B virus, HCV: Hepatitis C virus, HIV: human immunodeficiency virus, HSV: Herpes simplex virus, JEV: Japanese encephalitis virus, KSHV: Kaposi sarcoma-associated herpesvirus, 12/15-LOX:12/15-lipoxygenase, NRAMP: Natural Resistance-Associated Macrophage Proteins, ORF-3a: Open Reading Frame-3a, RSV: respiratory syncytial virus, SARS-CoV-2: severe acute respiratory syndrome-coronavirus-2, SARS-CoV: severe acute respiratory syndrome coronavirus, TfR1: transferrin receptor 1 protein, WNV: West Nile virus.

**Table 7 pathogens-13-00039-t007:** NRF2 agonists.

NRF2 Activator	Impact	Reference
4-OI	∙ ↓ viral replication of SARS-CoV-2.	[188]
DMF	∙ Inhibition of SARS-CoV-2 viral replication.	[188]
SFN	∙ Anti-SARS-CoV-2 properties through induction of NQO1 activation. ∙ Anti-viral properties against RSV. ∙ ↓ in viral load and IL-6 in influenza infection.	[37,189,190,194]
Curcumin	∙ Inhibition of influenza, parainfluenza, and RSV viral replication. ∙ ↓ in oxidative stress through HO-1 activation.	[51,191,192]
EGCG	∙ Anti-influenza activity through blockage of viral entry and subsequently viral replication.	[11]
Carbocistein	∙ ↓ in the expression of the nucleoprotein of influenza virus and thus viral replication.	[193]
BHA	∙ Severe ↓ in RSV-induced oxidative stress.	[168]

Abbreviations: BHA: butylated hydroxyanisole, DMF: dimethyl fumarate, EGCG: epigallocatechin gallate, HO-1: Heme Oxygenase 1, IL-6: Interleukin 6, NRF2: nuclear factor erythroid 2-related factor, 4-OI: 4-octyl itaconate, RSV: respiratory syncytial virus, SARS-CoV-2: severe acute respiratory syndrome coronavirus 2, SFN: Sulforaphane.

## Data Availability

Not applicable.

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
