# Peer review of "The Role of the NRF2 Pathway in the Pathogenesis of Viral Respiratory Infections"

_pathogens, 2023, doi:10.3390/pathogens13010039_

Round 1

Reviewer 1 Report

Comments and Suggestions for Authors

First of all, I would like to congratulate the authors for the comprehensive and complete review they have written. It is very pleasant to read and the readers can undertand clearly the importance of Nrf-2 pathway in the context of viral infections for the viruses cited. 

Tables 1-7 for example is a useful way to give the reader a comprehensive list of all impacts viral infections can have on the Nrf-2 pathway and the consequences on this pathway and on other inflamatory pathways.

I have few comments thought that I believe can improve the readability of figures shown in the review :

- Figure 1 : I would suggest to remove the background images behind the more detailed boxes, it doesn't give much more information and forces the authors to reduce the size of the important parts to fit the images.

- Figure 4 : The bottom right schematic is a bit too small and prevent the readers to be able to easily read what is written inside that schematic. I would suggest to adapt it and to summarize what is written if possible or to increase the font size.

some typos can be spotted through out the maniscript like in line 426 : NF-B instead of NF-kB.

Other than that, the maniscript is well written and has been a pleasure to read.

Reviewer 2 Report

Comments and Suggestions for Authors

Comments on the Quality of English Language

Needs some editing. 
